# Margin Discrepancy-based Adversarial Training for Multi-Domain Text Classification

## Abstract

Multi-domain text classification (MDTC) endeavors to harness available resources from correlated domains to enhance the classification accuracy of the target domain. Presently, most MDTC approaches that embrace adversarial training and the shared-private paradigm exhibit cutting-edge performance. Unfortunately, these methods face a non-negligible challenge: the absence of theoretical guarantees in the design of MDTC algorithms. The dearth of theoretical underpinning poses a substantial impediment to the advancement of MDTC algorithms. To tackle this problem, we first provide a theoretical analysis of MDTC by decomposing the MDTC task into multiple domain adaptation tasks. We incorporate the margin discrepancy as the measure of domain divergence and establish a new generalization bound based on Rademacher complexity. Subsequently, we propose a margin discrepancy-based adversarial training (MDAT) approach for MDTC, in accordance with our theoretical analysis. To validate the efficacy of the proposed MDAT method, we conduct empirical studies on two MDTC benchmarks. The experimental results demonstrate that our MDAT approach surpasses state-of-the-art baselines on both datasets.

## 1 Introduction

A fundamental assumption underlying supervised machine learning is that the training and test data should consist of independent and identically distributed (i.i.d.) samples drawn from the same distribution. Regrettably, it is common to encounter situations where there is a dearth of labeled data in the target domain, while an abundance of labeled data exists in related domains. Consequently, a classifier trained on one dataset is likely to exhibit poor performance on another dataset with a dissimilar distribution. Therefore, it is crucial to explore how to leverage the available resources from the related domains to enhance the classification accuracy in the target domain. Multi-domain text classification (MDTC) emerges as a specific machine learning task designed to tackle this very problem. In the MDTC setting, labeled data may be available for multiple domains; however, they are insufficient to train an effective classifier for one or more of these domains (Li & Zong, 2008; Wu & Guo, 2020).

One straightforward approach to perform MDTC in neural networks is to combine all labeled data from existing domains into a unified training set, disregarding the disparities across domains. This stream is referred to as the domain-agnostic method. However, it is well-known that text classification is a highly domain-dependent task, as the same word can convey different sentiments across various domains. For instance, consider the phrase "It runs fast" in the reviews of three products: Phone, Battery, and Car. Evidently, the term "**fast**" carries a positive sentiment when associated with Car and Phone, while expressing negative polarity in relation to Battery. Consequently, the domain-agnostic methods fall short in producing satisfactory results for MDTC. To take the domain difference into consideration, transfer learning is incorporated into MDTC to capture the interrelationships among different domains (Liu et al., 2017). By harnessing all available resources, including both labeled and unlabeled data, transfer learning empowers the enhancement of classification accuracy in the target domain (Pan & Yang, 2009).

Transfer learning-based MDTC methods, which employ diverse statistical matching techniques (Collobert & Weston, 2008; Wu & Huang, 2015) and discrepancy minimization methods (Liu et al., 2017; Chen & Cardie, 2018; Wu & Guo, 2020), have shown superiority over domain-agnostic approaches (Collobert & Weston, 2008; Wu & Huang, 2015; Liu et al., 2017; Chen & Cardie, 2018). Among these methods, those utilizing adversarial training and the shared-private paradigm achieve state-of-the-art performance. Drawing inspiration from the work of (Goodfellow et al., 2014; Ganin et al., 2016), adversarial training is employed to align different domains in the latent space, aiming to generate domain-invariant features that are both discriminative and transferable. Furthermore, (Bousmalis et al., 2016) proposes the shared-private paradigm in adversarial domain adaptation, illustrating that domain-specific knowledge can enhance the discriminability of the domain-invariant features.

Although many recent MDTC methods design sophisticated algorithms to achieve performance enhancements, only a limited number of studies have approached MDTC from a theoretical perspective. To date, only (Chen & Cardie, 2018) provides theoretical justifications for the convergence conditions of MDTC using two forms of f-divergence metrics (i.e., the least square loss and the negative log-likelihood loss). However, a comprehensive generalization bound for MDTC remains absent. Consequently, when designing an MDTC algorithm, we are at risk of lacking theoretical guarantees, given the significant gap that exists between theories and algorithms. This absence of theoretical assurance poses a significant obstacle to the advancement of MDTC techniques.

In this paper, we aim to bridge the aforementioned gap by conducting a comprehensive theoretical analysis of incorporating the margin discrepancy into adversarial MDTC. The margin discrepancy has been demonstrated to be a robust metric for measuring domain divergence in domain adaptation (Zhang et al., 2019), and we extend its application to the multi-domain scenario. Our analysis involves dividing the MDTC task into multiple domain adaptation tasks, incorporating the margin discrepancy into MDTC, and deriving a generalization bound based on Rademacher complexity for MDTC. Guided by these theoretical insights, we propose a margin discrepancy-based adversarial training (MDAT) method. Subsequently, we conduct a series of empirical studies to demonstrate the superior performance of our proposed MDAT method compared to state-of-the-art approaches on two MDTC benchmarks.

## 2 RELATED WORKS

### 2.1 DOMAIN ADAPTATION

Domain adaptation is proposed to transfer knowledge from a label-dense (source) domain to a label-scarce (target) domain (Pan & Yang, 2009). Earlier domain adaptation approaches employed normalization techniques and maximum mean discrepancy (MMD) to achieve this objective (Sun et al., 2016; Yan et al., 2017; Baktashmotlagh et al., 2014; Long et al., 2016). More recently, adversarial training has shown effectiveness in knowledge transfer (Ganin et al., 2016), and the adversarial domain adaptation methods can yield state-of-the-art performance (Farahani et al., 2021). Adversarial training was initially proposed for image generation (Goodfellow et al., 2014) and was later extended to align domains for learning domain-invariant features in domain adaptation (Ganin et al., 2016). In adversarial domain adaptation, a minimax optimization is employed between a domain discriminator and a feature extractor: the domain discriminator aims to distinguish between source and target features, while the feature extractor endeavors to confuse the domain discriminator. When adversarial training reaches its optimum, the learned features are considered domain-invariant, possessing both transferability and discriminability. Maximum classifier discrepancy (MCD) introduces a classifier-based adversarial training paradigm, where two classifiers are trained against a feature extractor to learn domain-invariant features. MCD optimizes the discrepancies between the outputs of the two classifiers in an adversarial manner, eliminating redundant hypothesis classes and yielding improvements (Saito et al., 2018). Margin Disparity Discrepancy (MDD) introduces the concept of margin discrepancy, which utilizes asymmetric margin loss to measure distribution divergence in domain adaptation (Zhang et al., 2019).

## 2.2 MULTI-DOMAIN TEXT CLASSIFICATION

The objective of multi-domain text classification (MDTC) is to utilize the available resources from existing domains to enhance the classification accuracy of the target domain (Li & Zong, 2008). Earlier MDTC approaches often trained multiple independent models per domain and aggregated their outputs for final predictions (Collobert & Weston, 2008; Liu et al., 2015; Wu & Huang, 2015). The most recent MDTC methods employ adversarial training and the shared-private paradigm (Liu et al., 2017; Chen & Cardie, 2018; Wu & Guo, 2020), which have demonstrated state-of-the-art performance. The shared-private paradigm emphasizes the significant contribution of domain-specific knowledge in enhancing the discriminability of domain-invariant features (Bousmalis et al., 2016). The adversarial multi-task learning for text classification (ASP-MTL) utilizes long short-term memory (LSTM) networks without attention as feature extractors and incorporates orthogonality constraints to encourage the shared and private feature extractors to capture distinct aspects of the inputs (Liu et al., 2017). The multinomial adversarial network (MAN) provides theoretical justification for the convergence conditions of MDTC, specifically regarding the least square loss and the negative log-likelihood loss (Chen & Cardie, 2018). The dual adversarial co-learning (DACL) method combines two forms of adversarial learning: (1) standard discriminator-based adversarial learning and (2) classifier-based adversarial learning, effectively guiding the feature extraction through adversarial alignment (Wu & Guo, 2020).

In contrast to previous adversarial MDTC methods that rely on combining multiple regularizers or complex network architectures for improved performance, our approach delves into the theoretical aspects of MDTC. The proposed method, known as the margin discrepancy-based adversarial training (MDAT), utilizes the margin discrepancy to capture the divergence between domains and guide the extraction of domain-invariant features. Furthermore, we derive an explicit generalization bound based on Rademacher complexity specifically for our MDAT method. To our knowledge, this is the first attempt to establish a generalization bound for MDTC.

## 3 PRELIMINARIES

### 3.1 LEARNING SETUP

In the context of supervised learning, the model undergoes training on a dataset denoted as $\widehat{D} = (\mathbf{x}_i, y_i)_{i=1}^n$ originating from $\mathcal{X} \times \mathcal{Y}$. Here, $\mathcal{X}$ represents the input space, while $\mathcal{Y}$ denotes the output space. Specifically, $\mathcal{Y}$ comprises the set $\{1, ..., k\}$ where $k$ signifies the total number of classes. The samples within the dataset $\widehat{D}$ are independent and identically distributed, drawn from the corresponding distribution $D$.

In the MDTC setting, let us consider $M$ distinct datasets denoted as $\{\widehat{D}_i\}_{i=1}^M$. Each $\widehat{D}_i$ comprises two components: a limited set of labeled samples $\mathbb{L}_i = \{(\mathbf{x}_j, y_j)\}_{j=1}^{l_i}$ and a substantial set of unlabeled samples $\mathbb{U}_i = \{\mathbf{x}_j\}_{j=1}^{u_i}$. Here, $l_i$ is the number of labeled samples and $u_i$ is the number of unlabeled samples. Each dataset $\widehat{D}_i$ encompasses a total of $n_i$ samples, where $n_i = l_i + u_i$. Notably, the samples within each dataset $\widehat{D}_i$ are independently and identically drawn from the distribution $D_i$. The primary goal of MDTC is to enhance system performance by effectively leveraging all available resources from the existing domains. The system performance is quantified as the average classification accuracy across the $M$ domains.

Adhering to the notations established by (Mohri et al., 2018), we examine the realm of classification within the hypothesis space $\mathcal{F}$ comprising scoring functions $f : \mathcal{X} \mapsto \mathbb{R}^{|\mathcal{Y}|} = \mathbb{R}^k$, where the outputs on each dimension signify the confidence score associated with the prediction. In MDTC, we are mainly dealing with the binary classification, i.e., $\mathbb{R}^{|\mathcal{Y}|} = \mathbb{R}^2$. The prediction corresponding to an instance $\mathbf{x}$ is the one yielding the highest score $f(\mathbf{x}, y)$. Consequently, a labeling function space $\mathcal{H}$ containing $h_f$ from $\mathcal{X}$ to $\mathcal{Y}$ can be induced:

$$h_f : \mathbf{x} \mapsto \arg\max_{y \in \mathcal{Y}} f(\mathbf{x}, y) \tag{1}$$

The expected error rate and the empirical error rate of a hypothesis class $h \in \mathcal{H}$ with respect to the distribution $D$ and its corresponding dataset $\widehat{D}$ are given by:

$$err_D(h) \triangleq \mathbb{E}_{(\mathbf{x},y)\sim D} \mathbb{1}[(h(\mathbf{x}) \neq y)] \tag{2}$$

$$err_{\widehat{D}}(h) \triangleq \mathbb{E}_{(\mathbf{x}_i,y_i)\sim \widehat{D}} \mathbb{1}[(h(\mathbf{x}_i) \neq y_i)] = \frac{1}{n} \sum_{i=1}^{n} \mathbb{1}[h(\mathbf{x}_i) \neq y_i] \tag{3}$$

where $\mathbb{1}[\mathbf{a}]$ is the indicator function, which is 1 if predicting $\mathbf{a}$ is true and 0 otherwise. For brevity, we only present the expected equations in the following.

## 3.2 ADVERSARIAL MULTI-DOMAIN TEXT CLASSIFICATION

Adversarial training has gained widespread utilization in aligning diverse feature distributions within the latent space to generate domain-invariant features in transfer learning, including domain adaptation (Ben-David et al., 2007; Ganin et al., 2016) and MDTC (Chen & Cardie, 2018; Wu & Guo, 2020). The domain adaptation has made significant theoretical advancements (Ben-David et al., 2007; 2010; Zhang et al., 2019), whereas MDTC still lacks comprehensive theoretical exploration. To date, only the multinomial adversarial networks (MANs) (Chen & Cardie, 2018) induce the convergence conditions for adversarial MDTC: Define the distribution of shared features $\mathbf{g}$ extracted from the distribution $D_i$ as follows:

$$P_i(\mathbf{g}) \triangleq P(\mathbf{g} = \mathcal{F}_s(\mathbf{x})|\mathbf{x} \in D_i) \tag{4}$$

Where $\mathcal{F}_s(\cdot)$ represents the shared feature embedding function and $P(\cdot)$ represents the probability. Then $P_1, P_2, ..., P_M$ are the $M$ shared feature distributions. Let $\bar{P} = \frac{\sum_{i=1}^{M} P_i}{M}$ be the centroid of the $M$ feature distributions. (Chen & Cardie, 2018) theoretically justify that the condition where an adversarial MDTC method reaches its optimum is $P_1 = P_2 = ... = P_M = \bar{P}$. Based on this observation, the main objective of MDTC can be regarded as minimizing the total divergence between each of the shared feature distributions of the $M$ domains and the centroid of the $M$ feature distributions.

## 3.3 DIVERGENCE MEASUREMENTS

In this section, we review the measurement of divergence for two different distributions in domain adaptation. In the seminal work of (Ben-David et al., 2007), given two hypothesis classes $h, h' \in \mathcal{H}$, the $\mathcal{H}\Delta\mathcal{H}$ divergence between two distributions $P$ and $Q$ is defined as:

$$d_{\mathcal{H}\Delta\mathcal{H}} = \sup_{h,h'\in\mathcal{H}} |\mathbb{E}_Q \mathbb{1}[h' \neq h] - \mathbb{E}_P \mathbb{1}[h' \neq h]| \tag{5}$$

(Mansour et al., 2009) extends the $\mathcal{H}\Delta\mathcal{H}$ divergence to general loss functions, leading to the discrepancy divergence:

$$dis_{\mathcal{L}} = \sup_{h,h'\in\mathcal{H}} |\mathbb{E}_Q \mathcal{L}(h', h) - \mathbb{E}_P \mathcal{L}(h', h)| \tag{6}$$

where $\mathcal{L}$ is a loss function. With these divergence measurements, generalization bounds of domain adaptation in terms of VC-dimension and Rademacher complexity are derived in (Ben-David et al., 2007; Mansour et al., 2009; Zhang et al., 2019).

### 3.4 Margin Discrepancy

The margin discrepancy is proposed as a means to quantify the domain divergence in domain adaptation, employing scoring functions and margin loss (Zhang et al., 2019). The utilization of margin loss allows for robust generalization performance between training samples and the classification surface in supervised classification (Lin, 2004; Koltchinskii et al., 2002). Define the margin of a hypothesis $f$ at a labeled sample $(\mathbf{x}, y)$ as:

$$\rho_f(\mathbf{x}, y) \triangleq \frac{1}{2}(f(\mathbf{x}, y) - \max_{y' \neq y} f(\mathbf{x}, y')) \tag{7}$$

Then the margin loss of a hypothesis $f$ with respect to the distribution $D$ can be defined as:

$$err_D^\rho(f) \triangleq \mathbb{E}_{(\mathbf{x},y) \sim D} \Phi_\rho \circ \rho_f(\mathbf{x}, y) \tag{8}$$

where $\circ$ denotes function composition, and $\Phi_\rho$ is defined as:

$$\Phi_\rho(\mathbf{x}) \triangleq \begin{cases} 0 & \rho \leq \mathbf{x} \\ 1 - \mathbf{x}/\rho & 0 \leq \mathbf{x} \leq \rho \\ 1 & \mathbf{x} \leq 0 \end{cases} \tag{9}$$

Given $h, h' \in \mathcal{H}$, the expected *0-1* disparity between $h$ and $h'$ with respect to the distribution $D$ can be defined as:

$$dis_D(h, h') \triangleq \mathbb{E}_D \mathbb{1}[h' \neq h] \tag{10}$$

Furthermore, given $h \in \mathcal{H}$, the *0-1* discrepancy induced by $h' \in \mathcal{H}$ between two dissimilar distributions $D_1$ and $D_2$ can be defined as:

$$d_{h,\mathcal{H}}(D_1, D_2) \triangleq \sup_{h' \in \mathcal{H}} (dis_{D_2}(h, h') - dis_{D_1}(h, h')) = \sup_{h' \in \mathcal{H}} (\mathbb{E}_{D_2} \mathbb{1}[h' \neq h] - \mathbb{E}_{D_1} \mathbb{1}[h' \neq h]) \tag{11}$$

By incorporating the margin loss to replace the *0-1* loss, the margin disparity can be induced as:

$$dis_D^\rho(f, f') \triangleq \mathbb{E}_{\mathbf{x} \sim D} \Phi_\rho \circ \rho_{f'}(\mathbf{x}, h_f(\mathbf{x})) \tag{12}$$

Where $f$ and $f'$ are scoring functions while $h_f$ and $h_{f'}$ are their labeling functions. Based on the margin disparity, the margin discrepancy can be formulated as:

$$d_{f,\mathcal{F}}^\rho(D_1, D_2) \triangleq \sup_{f' \in \mathcal{F}} (dis_{D_2}^\rho(f, f') - dis_{D_1}^\rho(f, f')) \tag{13}$$

The margin discrepancy is theoretically justified to be a well-defined discrepancy metric capable of effectively quantifying the divergence between two domains (Zhang et al., 2019).

## 4 Methodology

In this paper, we are driven by three compelling motivations: (1) The existence of strong connections between domain adaptation and MDTC; (2) The absence of a formal study on the margin discrepancy, derived from scoring functions and margin loss, in the multi-domain scenario; (3) To date, there is no work inducing a generalization bound for MDTC. Therefore, in this section, we commence by presenting a theoretical analysis that treats the MDTC task as $M$ different domain adaptation tasks between $\{P_i\}_{i=1}^M$ and $\bar{P}$, employing the margin discrepancy to encode the domain divergence, and deriving the generalization bound based on Rademacher complexity for MDTC. Subsequently, based on our theoretical findings, we propose the margin discrepancy-based adversarial training (MDAT) algorithm. We refer our readers to the Appendix for the proof details.

## 4.1 THEORETICAL ANALYSIS

According to (Zhang et al., 2019), the margin discrepancy is a well-defined discrepancy metric for domain adaptation due to the following proposition.

**Proposition 4.1.** *For any scoring function $f \in \mathcal{F}$,*

$$err_{D_T}(f) \leq err_{D_S}^{\rho}(f) + d_{f,\mathcal{F}}^{\rho}(D_T, D_S) + \lambda \tag{14}$$

*where $\lambda$ is a constant which is independent of $f$.*

In the above equation, $err_{D_S}^{\rho}(f)$ indicates the performance of $f$ on the source domain $D_S$ and the margin discrepancy $d_{f,\mathcal{F}}^{\rho}(D_T, D_S)$ bounds the performance gap caused by the domain shift between the source domain $D_S$ and the target domain $D_T$. $\lambda$ is determined by the practical problem and can be regarded as a small constant if the hypothesis space is sufficiently large. As the MDTC task can be intuitively decomposed into $M$ independent domain adaptation tasks between $\{D_i\}_{i=1}^{M}$ and $\bar{D}$, where $\bar{D} = \frac{\sum_{i=1}^{M} D_i}{M}$, we can extend the above learning bound to MDTC as:

$$err_{\bar{D}}(f) \leq \frac{1}{M} \sum_{i=1}^{M} [err_{D_i}^{\rho}(f) + d_{f,\mathcal{F}}^{\rho}(D_i, \bar{D})] + \lambda \tag{15}$$

The updated learning bound provides theoretical guarantees to advance MDTC by minimizing the domain divergence between each of $\{D_i\}_{i=1}^{M}$ and $\bar{D}$, and gives new standpoints to explore MDTC theoretically. To further induce a generalization bound for MDTC, we introduce the Rademacher complexity. The Rademacher complexity is widely used for deriving generalization bound in classification, it is a measurement of richness for a particular hypothesis space (Mohri et al., 2018).

**Definition 4.2.** (Rademacher Complexity) Assume $\mathcal{F}$ is the hypothesis space which maps $\mathcal{Z} = \mathcal{X} \times \mathcal{Y}$ to [a,b], and $\widehat{D} = \{z_1, z_2, ..., z_n\}$ denotes a dataset of size $n$ whose samples are drawn from the distribution $D$ over $\mathcal{Z}$. Then the empirical Rademacher complexity of $\mathcal{F}$ with regard to the dataset $\widehat{D}$ is defined as:

$$\widehat{\mathfrak{R}}_{\widehat{D}}(\mathcal{F}) \triangleq \mathbb{E}_{\sigma} \sup_{f \in \mathcal{F}} \frac{1}{n} \sum_{i=1}^{n} \sigma_i f(z_i) \tag{16}$$

where $\sigma_i$'s are independent uniform random variables taking values in $\{-1, +1\}$. The Rademacher complexity is:

$$\mathfrak{R}_{n,D}(\mathcal{F}) \triangleq \mathbb{E}_{\widehat{D} \sim D} \widehat{\mathfrak{R}}_{\widehat{D}}(\mathcal{F}) \tag{17}$$

Then, we estimate the margin discrepancy between two distributions $D_1$ and $D_2$ through finite samples using the Rademacher complexity.

**Lemma 4.3.** *For any $\delta > 0$, with probability $1 - 2\delta$,*

$$|d_{f,\mathcal{F}}^{\rho}(\widehat{D}_1, \widehat{D}_2) - d_{f,\mathcal{F}}^{\rho}(D_1, D_2)| \leq \frac{k}{\rho} \mathfrak{R}_{n_1, D_1}(\Pi_{\mathcal{H}}\mathcal{F}) + \frac{k}{\rho} \mathfrak{R}_{n_2, D_2}(\Pi_{\mathcal{H}}\mathcal{F}) + \sqrt{\frac{\log \frac{2}{\delta}}{2n_1}} + \sqrt{\frac{\log \frac{2}{\delta}}{2n_2}} \tag{18}$$

where $\Pi_{\mathcal{H}}\mathcal{F} = \{\mathbf{x} \mapsto f(\mathbf{x}, h_f(\mathbf{x})) | h_f \in \mathcal{H}, f \in \mathcal{F}\}$ is the scoring version of the symmetric difference hypothesis space $\mathcal{H}\Delta\mathcal{H}$ (Zhang et al., 2019), $n_1$ and $n_2$ are sizes of datasets $\widehat{D}_1$ and $\widehat{D}_2$, respectively. Combining proposition 4.1 with lemma 4.3, a new Rademacher complexity generalization bound for MDTC can be derived:

**Theorem 4.4.** *(Generalization Bound) For all $\delta > 0$, with probability $1 - 3\delta$,*

$$err_{\bar{D}}(f) \leq \frac{1}{M} \sum_{i=1}^{M} [err_{\widehat{D}_i}^{\rho}(f) + d_{f,\mathcal{F}}^{\rho}(D_i, \bar{D})] + \lambda + \frac{1}{M} \sum_{i=1}^{M} [\frac{8}{\rho} \mathfrak{R}_{n_i, D_i}(\Pi_1 \mathcal{F}) + \frac{2}{\rho} \mathfrak{R}_{n_i, D_i}(\Pi_{\mathcal{H}}\mathcal{F})$$

$$+ 2\sqrt{\frac{\log \frac{2}{\delta}}{2n_i}}] + \frac{2}{\rho} \mathfrak{R}_{\bar{n}, \bar{D}}(\Pi_{\mathcal{H}}\mathcal{F}) + \sqrt{\frac{\log \frac{2}{\delta}}{2\bar{n}}} \tag{19}$$

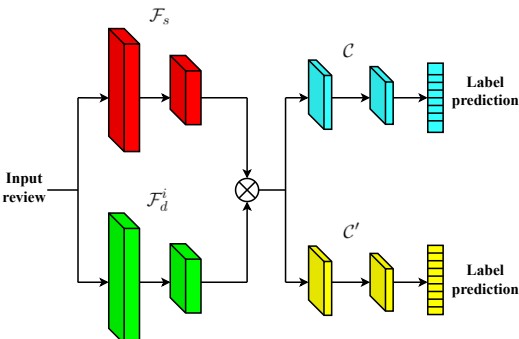

Figure 1: The architecture of the MDAT method.

where $\bar{n} = \frac{\sum_{i=1}^{M} n_i}{M}$. In essence, our work is a bold attempt to derive a generalization bound for MDTC, thereby endowing the field with theoretical underpinnings. Compared with the theoretical analysis provided in (Chen & Cardie, 2018), which only justifies the convergence conditions of MDTC, our theoretical findings are more informative.

## 4.2 MARGIN DISCREPANCY-BASED ADVERSARIAL TRAINING

Based on the above theoretical findings, we propose a margin discrepancy-based adversarial training (MDAT) method. As the margin discrepancy measures the domain divergence by restricting the hypothesis, we do not need the domain discriminator to distinguish features among multiple domains. Thus, we embrace the classifier-based adversarial training paradigm in our approach. The MDAT method consists of four components: a shared feature extractor $\mathcal{F}_s$, a set of domain-specific feature extractor $\{\mathcal{F}_d^i\}_{i=1}^{M}$, a classifier $\mathcal{C}$ and an auxiliary classifier $\mathcal{C}'$, the architecture is shown in Figure 1. The shared feature extractor $\mathcal{F}_s$ learns domain-invariant features that enhance classification performance across all domains, whereas each domain-specific feature extractor $\mathcal{F}_d^i$ captures domain-specific features from the $i$-th domain $\widehat{D}_i$ that solely contribute to classification within $\widehat{D}_i$. The auxiliary classifier $\mathcal{C}'$ complements the main classifier $\mathcal{C}$ by facilitating a minimax optimization against the shared feature extractor $\mathcal{F}_s$. The output of a (shared/domain-specific) feature extractor is a fixed-length vector. The classifiers take the concatenation of a shared feature vector and a domain-specific feature vector as input and produce the label probability.

According to Theorem 4.4, the main objective of MDTC is minimizing the error rates on all datasets $\{\widehat{D}_i\}_{i=1}^{M}$, and the total divergence between each of the distributions of the $M$ domains and the centroid of the $M$ distributions. We thus need to solve the following minimization problem in the hypothesis space $\mathcal{F}$:

$$\min_{f \in \mathcal{F}} \sum_{i=1}^{M} [err_{D_i}^{\rho}(f) + d_{f,\mathcal{F}}^{\rho}(D_i, \bar{D})] \tag{20}$$

Minimizing the margin discrepancy is equivalent to performing a minimax optimization since the margin discrepancy is defined as the supremum over the hypothesis space $\mathcal{F}$ (Zhang et al., 2019). Denote the feature embedding function as $\phi$ and the classifying function as $\omega$, where $f$ can be decomposed as $f = \omega \circ \phi$. Applying $\phi$ and $\omega$ to the $M$ domains, the overall optimization problem can be defined as follows:

$$\min_{\omega, \phi} \sum_{i=1}^{M} [err_{\phi(\widehat{D}_i)}^{\rho}(\omega) + dis_{\phi(\widehat{D})}^{\rho}(\omega, \omega^*) - dis_{\phi(\widehat{D}_i)}^{\rho}(\omega, \omega^*)]$$

$$\omega^* = \max_{\omega'} \sum_{i=1}^{M} [dis_{\phi(\widehat{D})}^{\rho}(\omega, \omega') - dis_{\phi(\widehat{D}_i)}^{\rho}(\omega, \omega')] \tag{21}$$

where $\omega'$ is the classifying function of the auxiliary classifier $\mathcal{C}'$. By adversarially optimizing the discrepancy formed by the outputs of the two classifiers, the redundant hypothesis classes can be ruled out such that the domain-invariant features can be learned.

As the margin loss can cause gradient vanishing in stochastic gradient descent in practice (Koltchinskii et al., 2002), and thus cannot be optimized efficiently for representation learning that significantly relies on gradient propagation, we need to use other loss functions to approximate the margin discrepancy. For convenience, we use the cross-entropy loss to rewrite the above optimization problem.

$$\min_{\mathcal{F}_s, \mathcal{F}_d^i, \mathcal{C}} \max_{\mathcal{C}'} \mathcal{J}_C + \alpha \mathcal{J}_D \tag{22}$$

$$\mathcal{J}_C = -\sum_{i=1}^{M} \mathbb{E}_{(\mathbf{x}, y) \sim \mathbb{L}_i} \log(\mathcal{C}_y[\mathcal{F}^i(\mathbf{x})]) \tag{23}$$

$$\mathcal{J}_D = \sum_{i=1}^{M} [\beta \mathbb{E}_{\mathbf{x} \sim \mathbb{L}_i \cup \mathbb{U}_i} \log(\mathcal{C}'_{\sigma[\mathcal{F}^i(\mathbf{x})]}[\mathcal{F}^i(\mathbf{x})]) + \mathbb{E}_{\mathbf{x} \sim \mathbb{L}_i \cup \mathbb{U}_i} \log(1 - \mathcal{C}'_{\sigma[\mathcal{F}^i(\mathbf{x})]}[\mathcal{F}^i(\mathbf{x})])] \tag{24}$$

Where $\beta = \exp(\rho)$ is designed to attain the margin, $\mathcal{F}^i(\mathbf{x}) = [\mathcal{F}_s(\mathbf{x}), \mathcal{F}_d^i(\mathbf{x})]$ is the input of the classifiers where $[\cdot, \cdot]$ is the concatenation of two vectors. $\mathcal{C}_y([\mathcal{F}^i(\mathbf{x})])$ and $\mathcal{C}'_y([\mathcal{F}^i(\mathbf{x})])$ indicate the prediction probability of the given sample $\mathbf{x}$ belonging to the label $y$ by the classifier $\mathcal{C}$ and $\mathcal{C}'$, respectively. $\sigma(\cdot)$ represents the labelling function of the classifier $\mathcal{C}$. $\alpha$ is the trade-off hyperparameter balancing two competitive terms. By performing the minimax optimization in MDAT, we shall see that it will lead to $\phi(\widehat{D}_1) \approx \phi(\widehat{D}_2) \approx ... \approx \phi(\widehat{D}_M) \approx \phi(\widehat{D})$. The detailed training procedure is available in the Appendix

## 5 EXPERIMENTS

### 5.1 EXPERIMENTAL SETTINGS

**Dataset** We conduct experiments on two MDTC benchmarks: the Amazon review dataset (Blitzer et al., 2007) and the FDU-MTL dataset (Liu et al., 2017). The Amazon review dataset comprises 4 domains: books, dvds, electronics, and kitchen. Each domain has 2,000 samples with binary labels. All data have underwent preprocessing, resulting in a bag-of-features representation (including unigrams and bigrams), which does not preserve word order or grammar information. Consequently, the use of powerful feature extractors such as convolutional neural networks (CNNs), recurrent neural networks (RNNs), or long short-term memory (LSTM) is not viable. We hence follow (Chen & Cardie, 2018) to employ a multiple layer perceptron (MLP) as the feature extractor and represent each review as a 5000-dimensional vector whose values are raw counts of the features.

Given the limitations of the Amazon review dataset, such as its small number of domains and the conversion of reviews into a bag-of-features representation containing only unigrams and bigrams, we aim to further assess the efficacy of the MDAT method. For this purpose, we perform experiments on the FDU-MTL dataset (Liu et al., 2017), which comprises raw text data. Unlike the Amazon dataset, the FDU-MTL dataset allows us to leverage powerful neural networks as feature extractors and process the data from its original form. This dataset encompasses 16 domains, including books, electronics, DVDs, kitchen, apparel, camera, health, music, toys, video, baby, magazine, software, sport, IMDB, and MR. Detailed statistics regarding the FDU-MTL dataset and the implementation details can be found in the Appendix.

**Comparison Methods** We compare the MDAT method with a number of state-of-the-art methods: the multi-task convolutional neural network (MT-CNN) (Collobert & Weston, 2008), the multi-task deep neural network (MT-DNN) (Liu et al., 2015), the collaborative multi-domain text classification

Table 1: MDTC classification accuracies on the Amazon review dataset.

| Domain | CMSC-LS | CMSC-SVM | CMSC-Log | MAN-L2 | MAN-NLL | DACL | MDAT(Proposed) |
|--------|---------|----------|----------|--------|---------|------|----------------|
| Books  | 82.10   | 82.26    | 81.81    | 82.46  | 82.98   | 83.45| **84.33 $\pm$ 0.16** |
| DVD    | 82.40   | 83.48    | 83.73    | 83.98  | 84.03   | 85.50| **86.07 $\pm$ 0.11** |
| Electr.| 86.12   | 86.76    | 86.67    | 87.22  | 87.06   | 87.40| **88.74 $\pm$ 0.06** |
| Kit.   | 87.56   | 88.20    | 88.23    | 88.53  | 88.57   | 90.00| **90.56 $\pm$ 0.13** |
| AVG    | 84.55   | 85.18    | 85.11    | 85.55  | 85.66   | 86.59| **87.43 $\pm$ 0.08** |

Table 2: MDTC classification accuracies on the FDU-MTL dataset.

| Domain | MT-CNN | MT-DNN | ASP-MTL | MAN-L2 | MAN-NLL | DACL | MDAT(Proposed) |
|--------|--------|--------|---------|--------|---------|------|----------------|
| books       | 84.5 | 82.2 | 84.0 | 87.6 | 86.8 | 87.5   | **88.1 $\pm$ 0.2** |
| electronics | 83.2 | 88.3 | 86.8 | 87.4 | 88.8 | **90.3** | 88.8±0.6 |
| dvd         | 84.0 | 84.2 | 85.5 | 88.1 | 88.6 | 89.8   | **91.0 $\pm$ 0.4** |
| kitchen     | 83.2 | 80.7 | 86.2 | 89.8 | 89.9 | 91.5   | **92.1 $\pm$ 0.4** |
| apparel     | 83.7 | 85.0 | 87.0 | 87.6 | 87.6 | 89.5   | **90.3 $\pm$ 0.5** |
| camera      | 86.0 | 86.2 | 89.2 | 91.4 | 90.7 | 91.5   | **92.2 $\pm$ 0.4** |
| health      | 87.2 | 85.7 | 88.2 | 89.8 | 89.4 | **90.5** | 89.8±0.2 |
| music       | 83.7 | 84.7 | 82.5 | 85.9 | 85.5 | 86.3   | **87.5 $\pm$ 0.1** |
| toys        | 89.2 | 87.7 | 88.0 | 90.0 | 90.4 | 91.3   | **91.4 $\pm$ 0.4** |
| video       | 81.5 | 85.0 | 84.5 | 89.5 | 89.6 | 88.5   | **90.7 $\pm$ 0.6** |
| baby        | 87.7 | 88.0 | 88.2 | 90.0 | 90.2 | **92.0** | 90.0±0.3 |
| magazine    | 87.7 | 89.5 | 92.2 | 92.5 | 92.9 | 93.8   | **94.2 $\pm$ 0.2** |
| software    | 86.5 | 85.7 | 87.2 | 90.4 | 90.9 | 90.5   | **91.3 $\pm$ 0.5** |
| sports      | 84.0 | 83.2 | 85.7 | 89.0 | 89.0 | 89.3   | **89.7 $\pm$ 0.3** |
| IMDb        | 86.2 | 83.2 | 85.5 | 86.6 | 87.0 | 87.3   | **89.4 $\pm$ 0.1** |
| MR          | 74.5 | 75.5 | 76.7 | 76.1 | 76.7 | 76.0   | **77.0 $\pm$ 0.5** |
| AVG         | 84.5 | 84.3 | 86.1 | 88.2 | 88.4 | 89.1   | **89.6 $\pm$ 0.1** |

with the least-square loss (CMSC-L2), the hinge loss (CMSC-SVM), and the log loss (CMSC-log) (Wu & Huang, 2015), the adversarial multi-task learning for text classification (ASP-MTL) (Liu et al., 2017), the multinomial adversarial network with the least-square loss (MAN-L2) and the negative log-likelihood loss (MAN-NLL), and the dual adversarial co-learning method (DACL) (Wu & Guo, 2020).

## 5.2 RESULTS

All experiments are reported based on 5 random trials. The experimental results on the Amazon review dataset and FDU-MTL dataset are presented in Table 1 and Table 2, respectively. From Table 1, we observe that the MDAT method can outperform other baselines not only in terms of average accuracy, but also on each individual domain. In particular, our MDAT can reach the average accuracy of $87.43\%$, outperforming the second-best approach DACL method by $0.84\%$. For the experimental results on the FDU-MTL dataset, as shown in Table 2, it can be noted that our proposed MDAT method can outperform baselines on 13 out of 16 domains. In average classification accuracy, our MDAT method obtains the best performance of $89.6\%$. The experimental results on two MDTC benchmarks both validate the effectiveness of our model. The ablation study, parameter sensitivity analysis, and extensive experiments on multi-source unsupervised domain adaptation are presented in the Appendix to provide more insights into our MDAT method.

## 6 CONCLUSION

In this paper, we first present a theoretical MDTC analysis of separating the MDTC task into multiple independent domain adaptation tasks, incorporating the margin discrepancy to guide the adversarial alignment, and deriving a generalization bound based on Rademacher complexity for MDTC. To the best of our knowledge, our work is the first attempt to provide a generalization bound for MDTC. Based on our theoretical findings, we propose a margin discrepancy-based adversarial training (MDAT) method. The experimental results show that our MDAT method is an effective method with strong theoretical guarantees, outperforming state-of-the-art MDTC methods on two benchmarks.

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

## A    GENERALIZATION BOUNDS WITH MARGIN DISCREPANCY

**Theorem A.1.** *(Proposition 3.3 of (Zhang et al., 2019) & Proposition 4.1 in the main content) For any scoring function $f \in \mathcal{F}$,*

$$err_{D_T}(f) \leq err^\rho_{D_S}(f) + d^\rho_{f,\mathcal{F}}(D_T, D_S) + \lambda$$

*where $\lambda = \lambda(\rho, \mathcal{F}, D_S, D_T)$ is a constant independent of $f$.*

**Definition A.2.** (Definition 3.4 of (Zhang et al., 2019)) Given a hypothesis space $\mathcal{F}$ and its corresponding labeling space $\mathcal{H}$, $\Pi_\mathcal{H}\mathcal{F}$ is defined as

$$\Pi_\mathcal{H}\mathcal{F} = \{x \mapsto f(x, h_f(x)) | h_f \in \mathcal{H}, f \in \mathcal{F}\}$$

(Galbis & Maestre, 2012) presents a geometric interpretation of the set $\Pi_\mathcal{H}\mathcal{F}$. Assume that $\mathcal{X}$ is a manifold, we can obtain a vector bundle $\mathcal{B}$ by assigning a vector space $\mathbb{R}^k$ to each sample in $\mathcal{X}$. By considering the values of $\mathcal{H}$ as one-hot vectors in $\mathbb{R}^k$, $\mathcal{H}$ and $\mathcal{F}$ are both sets of sections of $\mathcal{B}$ containing vector fields. $\Pi_\mathcal{H}\mathcal{F}$ can be regarded as the space of inner products of vector fields with respect to $\mathcal{H}$ and $\mathcal{F}$:

$$\Pi_\mathcal{H}\mathcal{F} = \langle \mathcal{H}, \mathcal{F} \rangle = \{\langle h, f \rangle | h \in \mathcal{H}, f \in \mathcal{F}\}$$

**Lemma A.3.** *(A modified version of Theorem 8.1, (Mohri et al., 2018)). Suppose $\mathcal{F} \subseteq \mathbb{R}^{\mathcal{X} \times \mathcal{Y}}$ is the hypothesis set of scoring functions with $\mathcal{Y} = \{1, 2, ..., k\}$. Let*

$$\Pi_1\mathcal{F} \triangleq \{x \mapsto f(x, y) | y \in \mathcal{Y}, f \in \mathcal{F}\}$$

*Fix $\rho > 0$. Then for any $\delta > 0$, with probability at least $1 - \delta$, the following holds for all $f \in \mathcal{F}$:*

$$|err^\rho_D(f) - err^\rho_{\widehat{D}}(f)| \leq \frac{2k^2}{\rho}\mathfrak{R}_{n,D}(\Pi_1\mathcal{F}) + \sqrt{\frac{\log\frac{2}{\delta}}{2n}}$$

A simple corollary of this lemma is the margin bound for binary classification.

$$err_D(f) \leq err^\rho_D(f) \leq err^\rho_{\widehat{D}}(f) + \frac{8}{\rho}\mathfrak{R}_{n,D}(\Pi_1\mathcal{F}) + \sqrt{\frac{\log\frac{2}{\delta}}{2n}} \tag{25}$$

**Lemma A.4.** *(Talagrand's lemma, (Talagrand, 2014);(Mohri et al., 2018)). Let $\Phi : \mathbb{R} \mapsto \mathbb{R}$ be an $\ell$-Lipschitz. Then for any hypothesis space $\mathcal{F}$ of real-valued functions, and any sample $D$ of size n, the following inequality holds:*

$$\mathfrak{R}_{\widehat{D}}(\Phi \circ \mathcal{F}) \leq \ell\mathfrak{R}_{\widehat{D}}(\mathcal{F})$$

**Lemma A.5.** *(Lemma 8.1 of (Mohri et al., 2018)). Let $\mathcal{F}_1, ..., \mathcal{F}_k$ be k hypothesis sets in $\mathbb{R}^\mathcal{X}$, $k > 1$. $\mathcal{G} = \{\max\{f_1, ..., f_k\} : f_i \in \mathcal{F}, i \in \{1, ..., k\}\}$, then for any dataset $\widehat{D}$ of size n, we have:*

$$\widehat{\mathfrak{R}}_{\widehat{D}}(\mathcal{G}) \leq \sum_{i=1}^k \widehat{\mathfrak{R}}_{\widehat{D}}(\mathcal{F}_i)$$

**Theorem A.6.** *(Lemma 3.6 of (Zhang et al., 2019) & Lemma 4.3 in the main content). Let $\mathcal{F} \in \mathbb{R}^{\mathcal{X} \times \mathcal{Y}}$ is a hypothesis space. Let $\mathcal{H}$ be the set of classifiers (mapping $\mathcal{X}$ to $\mathcal{Y}$) corresponding to $\mathcal{F}$. For any $\delta > 0$, with probability $1 - 2\delta$, the following holds simultaneously for any scoring function $f$,*

$$|d^\rho_{f,\mathcal{F}}(\widehat{D}_1, \widehat{D}_2) - d^\rho_{f,\mathcal{F}}(D_1, D_2)| \leq \frac{k}{\rho}\mathfrak{R}_{n_1,D_1}(\Pi_\mathcal{H}\mathcal{F}) \quad + \frac{k}{\rho}\mathfrak{R}_{n_2,D_2}(\Pi_\mathcal{H}\mathcal{F}) + \sqrt{\frac{\log\frac{2}{\delta}}{2n_1}} + \sqrt{\frac{\log\frac{2}{\delta}}{2n_2}}.$$

**Theorem A.7.** *(Theorem 4.4 in the main content). For any $\delta > 0$, with probability $1 - 3\delta$, we have the following uniform generalization bound for all scoring functions $f$:*

$$err_{\bar{D}}(f) \leq \frac{1}{M} \sum_{i=1}^{M} [err^{\rho}_{\widehat{D}_i}(f) + d^{\rho}_{f,\mathcal{F}}(D_i, \bar{D})] + \lambda + \frac{1}{M} \sum_{i=1}^{M} [\frac{8}{\rho} \mathfrak{R}_{n_i,D_i}(\Pi_1 \mathcal{F}) + \frac{2}{\rho} \mathfrak{R}_{n_i,D_i}(\Pi_{\mathcal{H}} \mathcal{F})$$

$$+ 2\sqrt{\frac{\log \frac{2}{\delta}}{2n_i}}] + \frac{2}{\rho} \mathfrak{R}_{\bar{n},\bar{D}}(\Pi_{\mathcal{H}} \mathcal{F}) + \sqrt{\frac{\log \frac{2}{\delta}}{2\bar{n}}}$$

*Before obtaining this theorem, we combine Theorem A.1, Equation 25 and Theorem A.6 to get:*

$$err_{\bar{D}}(f) \leq err^{\rho}_{\widehat{D}_i}(f) + d^{\rho}_{f,\mathcal{F}}(\widehat{D}_i, \widehat{\bar{D}}) + \lambda + \frac{8}{\rho} \mathfrak{R}_{n_i,D_i}(\Pi_1 \mathcal{F}) + \frac{2}{\rho} \mathfrak{R}_{n_i,D_i}(\Pi_{\mathcal{H}} \mathcal{F})$$

$$+ 2\sqrt{\frac{\log \frac{2}{\delta}}{2n_i}} + \frac{2}{\rho} \mathfrak{R}_{\bar{n},\bar{D}}(\Pi_{\mathcal{H}} \mathcal{F}) + \sqrt{\frac{\log \frac{2}{\delta}}{2\bar{n}}}$$

By simple summation, we can get

$$M \times err_{\bar{D}}(f) \leq \sum_{i=1}^{M} [err^{\rho}_{\widehat{D}_i}(f) + d^{\rho}_{f,\mathcal{F}}(\widehat{D}_i, \widehat{\bar{D}}) + \lambda + \frac{8}{\rho} \mathfrak{R}_{n_i,D_i}(\Pi_1 \mathcal{F}) + \frac{2}{\rho} \mathfrak{R}_{n_i,D_i}(\Pi_{\mathcal{H}} \mathcal{F})$$

$$+ 2\sqrt{\frac{\log \frac{2}{\delta}}{2n_i}} + \frac{2}{\rho} \mathfrak{R}_{\bar{n},\bar{D}}(\Pi_{\mathcal{H}} \mathcal{F}) + \sqrt{\frac{\log \frac{2}{\delta}}{2\bar{n}}}]$$

Then we conclude the Theorem A.7.

## B  TRAINING PROCEDURE

---
**Algorithm 1** MDAT training algorithm

---
1: **Input:** labeled data $\mathbb{L}_i$ and unlabeled data $\mathbb{U}_i$ in $M$ domains; hyperparameters: $\alpha$ and $\beta$
2: **for** number of training iterations **do**
3:     Sample labeled mini-batches from the multiple domains $B^{\ell} = \{B^{\ell}_1, \cdots, B^{\ell}_M\}$.
4:     Sample unlabeled mini-batches from the multiple domains $B^u = \{B^u_1, \cdots, B^u_M\}$.
5:     Calculate $loss = \mathcal{J}_C + \alpha \mathcal{J}_D$ on $B^{\ell}$ and $B^u$;
        Update $\mathcal{F}_s$, $\{\mathcal{F}^i_d\}_{i=1}^{M}$, and $\mathcal{C}$ by descending along gradients $\nabla loss$.
6:     Calculate $l_D = \mathcal{J}_D$ on $B^{\ell}$ and $B^u$;
        Update $\mathcal{C}'$ by ascending along gradients $\nabla l_D$.
7: **end for**

---

The MDAT method is trained using mini-batch stochastic gradient descent (SGD) in an alternating fashion (Ganin et al., 2016). In each iteration, the input is passed through both the shared feature extractor and the corresponding domain-specific feature extractor to obtain feature representations. The resulting features, obtained by concatenating the shared and domain-specific features, are then fed into the two classifiers to generate predictions. The specific training procedure can be summarized in two steps: (1) Calculate $loss = \mathcal{J}_C + \alpha \mathcal{J}_D$ on both labeled and unlabeled data to update the parameters of $\mathcal{F}_s$, $\{\mathcal{F}^i_d\}_{i=1}^{M}$, and $\mathcal{C}$ by descending along the gradients of $\nabla loss$; (2) Calculate $l_D = \mathcal{J}_D$ on both labeled and unlabeled data to update the parameters of $\mathcal{C}'$ by ascending along the gradients of $\nabla l_D$. The MDAT method has two hyperparameters, $\alpha$ and $\beta$. $\alpha$ is used to balance different loss functions, while $\beta$ is designed to control the margin. In our experiments, we fix $\alpha = 0.5$ and $\beta = 4$. Please refer to Algorithm 1 for the detailed algorithmic description.

Table 3: Statistics of the Amazon review dataset

| Domain | Labeled | Unlabeled | Class. |
|--------|---------|-----------|--------|
| Books | 2000 | 4465 | 2 |
| Electronics | 2000 | 3586 | 2 |
| DVD | 2000 | 568 | 2 |
| Kitchen | 2000 | 5945 | 2 |

Table 4: Statistics of the FDU-MTL dataset

| Domain | Train | Dev. | Test | Unlabeled | Avg. L | Vocab. |
|--------|-------|------|------|-----------|--------|--------|
| Books | 1400 | 200 | 400 | 2000 | 159 | 62K |
| Electronics | 1398 | 200 | 400 | 2000 | 101 | 30K |
| DVD | 1400 | 200 | 400 | 2000 | 173 | 69K |
| Kitchen | 1400 | 200 | 400 | 2000 | 89 | 28K |
| Apparel | 1400 | 200 | 400 | 2000 | 57 | 21K |
| Camera | 1397 | 200 | 400 | 2000 | 130 | 26K |
| Health | 1400 | 200 | 400 | 2000 | 81 | 26K |
| Music | 1400 | 200 | 400 | 2000 | 136 | 60K |
| Toys | 1400 | 200 | 400 | 2000 | 90 | 28K |
| Video | 1400 | 200 | 400 | 2000 | 156 | 57K |
| Baby | 1300 | 200 | 400 | 2000 | 104 | 26K |
| Magazine | 1370 | 200 | 400 | 2000 | 117 | 30K |
| Software | 1315 | 200 | 400 | 475 | 129 | 26K |
| Sports | 1400 | 200 | 400 | 2000 | 94 | 30K |
| IMDB | 1400 | 200 | 400 | 2000 | 269 | 44K |
| MR | 1400 | 200 | 400 | 2000 | 21 | 12K |

## C  DATASETS

We conduct experiments on two MDTC benchmarks: the Amazon review dataset (Blitzer et al., 2007) and the FDU-MTL dataset (Liu et al., 2017). The Amazon review dataset comprises four distinct product review domains: books, DVDs, electronics, and kitchen. The data within the Amazon review dataset has undergone preprocessing, resulting in a bag-of-features that includes unigrams and bigrams, thereby omitting any word order information. On the other hand, the FDU-MTL dataset poses a more challenging task as it encompasses 16 diverse domains, including books, electronics, DVDs, kitchen, apparel, camera, health, music, toys, video, baby, magazine, software, sport, IMDB, and MR. Among these, the first 14 domains pertain to product reviews, while the last 2 domains specifically focus on movie reviews. The data within the FDU-MTL dataset consists of raw text data, having solely undergone tokenization using the Stanford Tokenizer (Manning et al., 2014). For further details regarding these datasets, please refer to Table 3 and 4.

## D  IMPLEMENTATION DETAILS

All experiments are conducted using Pytorch. To ensure fair comparisons, we adhere to the standard MDTC evaluating protocol (Chen & Cardie, 2018) and ensure that all baselines adopt the standard partitions of the datasets. Thus, we conveniently cite the experimental results from (Chen & Cardie, 2018; Wu & Guo, 2020). The MDAT has two hyperparameters: $\alpha$ and $\beta$, we fix $\alpha = 0.5$ and $\beta = 4$. The adam optimizer (Kingma & Ba, 2014) with a learning rate of 0.0001 is used for training. The batch size is 8. The dimension of the output of the shared feature extractor is 128 while 64 for the domain-specific feature extractor. The dropout rate for each component is set to 0.4. The classifiers adopt the MLP architecture with one hidden layer, the dimension of the hidden layer is the same size as its input, i.e., $128 + 64$. ReLU serves as the activation function. We report the performance of the main classifier.

When conducting the experiments on the Amazon review dataset, following (Chen & Cardie, 2018), a 5-way cross-validation is performed and the 5-fold average test accuracy is reported. MLP with two hidden layers is used as the feature extractor, the dimensions of the input and two hidden layers are 5,000, 1,000, and 500 respectively. For the experiments on the FDU-MTL dataset, we adopt a CNN as the feature extractor, aligning with the usage of CNN in recent baselines like MAN (Chen & Cardie, 2018) and DACL (Wu & Guo, 2020). To facilitate fair comparisons, we employ CNNs with a single convolutional layer as feature extractors. Our CNN feature extractors use different kernel

Table 5: Ablation Study on the Amazon review dataset

| Domain | Books | Dvds | Elec. | Kit. | Avg. |
|---|---|---|---|---|---|
| MDAT | 84.33 | 86.07 | 88.74 | 90.56 | 87.43 |
| MDAT w $\ell_1$ | 83.45 | 85.10 | 86.30 | 89.15 | 85.86 |

sizes (3, 4, 5), and the number of kernels is 200. The input of the CNN is a 100-dimensional vector, obtained by using word2vec (Mikolov et al., 2013), for each word in the input sequence.

# E   ADDITIONAL EXPERIMENTS

## E.1   ABLATION STUDY

To ensure that the performance improvements achieved by our MDAT method primarily arise from the margin discrepancy rather than the classifier-based adversarial training paradigm, we explore a variant: MDAT w $\ell_1$. In this variant, inspired by (Saito et al., 2018), we incorporate the $\ell_1$ norm to quantify the discrepancy formed by the outputs of the two classifiers. We also employ the three-step training procedure (Saito et al., 2018) to optimize the model parameters. The updated loss functions, denoted as $\mathcal{J}'_C$ and $\mathcal{J}'_D$, can be formulated as follows:

$$\mathcal{J}'_C = -\sum_{i=1}^{M} \mathbb{E}_{\mathbf{x} \sim \mathbb{L}_i} [\log(\mathcal{C}_y[\mathcal{F}^i(\mathbf{x})]) + \log(\mathcal{C}'_y[\mathcal{F}^i(\mathbf{x})])] \tag{26}$$

$$\mathcal{J}'_D = \sum_{i=1}^{M} \mathbb{E}_{\mathbf{x} \sim \mathbb{L}_i \cup \mathbb{U}_i} |\mathcal{C}[\mathcal{F}^i(\mathbf{x})] - \mathcal{C}'[\mathcal{F}^i(\mathbf{x})]| \tag{27}$$

where $\mathcal{C}[\cdot]$ and $\mathcal{C}'[\cdot]$ represent the softmax vector generated by $\mathcal{C}$ and $\mathcal{C}'$, respectively. $|\cdot|$ denotes the $\ell_1$ norm function. The three-step training fashion (Saito et al., 2018) is defined as (1) Calculate $l_C = \mathcal{J}'_C$ on the labeled data and update the parameters of $\mathcal{F}_s$, $\{\mathcal{F}^i_d\}_{i=1}^{M}$, $\mathcal{C}$ and $\mathcal{C}'$ by descending along the gradients of $\nabla l_C$; (2) Calculate $loss = \mathcal{J}'_C + \alpha\mathcal{J}'_D$ on both labeled and unlabeled data, fix the parameters of $\mathcal{C}$ and $\mathcal{C}'$, and update the parameters of $\mathcal{F}_s$ and $\{\mathcal{F}^i_d\}_{i=1}^{M}$ by descending along the gradients of $\nabla loss$; (3) Calculate $l_D = \mathcal{J}'_D$ on both labeled and unlabeled data, fix the parameters of $\mathcal{F}_s$ and $\{\mathcal{F}^i_d\}_{i=1}^{M}$, and update the parameters of $\mathcal{C}$ and $\mathcal{C}'$ by ascending along the gradients of $\nabla l_D$. We conduct the ablation study on the Amazon review dataset and the hyperparameter $\alpha$ is also fixed as $\alpha = 0.5$. The comparison results are shown in Table 5, we can observe that our MDAT method can not only outperform the variant on all four domains, but also beat the variant by 1.57% in terms of the average classification accuracy.

## E.2   PARAMETER SENSITIVITY ANALYSIS

The proposed MDAT approach incorporates two hyperparameters: $\alpha$ and $\beta$. $\alpha$ serves the purpose of balancing the two competing terms within the minimax optimization, while $\beta$ attains the margin. To analyze the sensitivity of these two hyperparameters, we perform experiments on the Amazon review dataset. Initially, we fix $\alpha = 0.5$ and conduct experiments using various $\beta$ values ranging from $\{1, 2, 3, 4, 5\}$. Subsequently, we fix $\beta = 4$ and carry out experiments with different $\alpha$ values in the range of $\{0.01, 0.1, 0.5, 1.0, 2.0\}$. Figure 2 presents the experimental results. Remarkably, the system performance is not significantly influenced by $\alpha$, thus we set $\alpha = 0.5$ for all our experiments. Regarding $\beta$, the average classification accuracy exhibits rapid improvement as $\beta$ increases from 1 to 4. However, further increments of $\beta$ lead to performance degradation. These findings underscore the importance of the margin in our MDAT method, emphasizing the need for an appropriate $\beta$ value to yield better results.

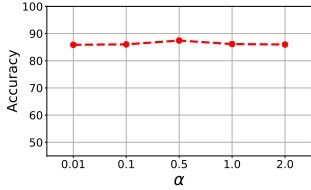 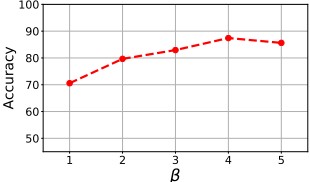

Figure 2: Parameter sensitivity analysis on the Amazon review dataset

Table 6: Multi-source unsupervised domain adaptation results on the Amazon review dataset.

| Domain | MLP | mSDA | DANN | MDAN | MAN-L2 | MAN-NLL | DACL | MDAT(Proposed) |
|--------|-----|------|------|------|--------|---------|------|----------------|
| Books | 76.55 | 76.98 | 77.89 | 78.63 | 78.45 | 77.78 | 80.22 | **80.76** |
| DVD | 75.88 | 78.61 | 78.86 | 77.97 | 81.57 | 82.74 | 82.96 | **83.41** |
| Elec. | 84.60 | 81.98 | 84.91 | 85.34 | 83.37 | 83.75 | 84.90 | **85.27** |
| Kit. | 85.45 | 84.26 | 86.39 | 86.26 | 85.57 | 86.41 | 86.75 | **86.98** |
| AVG | 80.46 | 80.46 | 82.01 | 82.72 | 82.24 | 82.67 | 83.71 | **84.11** |

### E.3 MULTI-SOURCE UNSUPERVISED DOMAIN ADAPTATION

In real-world scenarios, it is conceivable that the target domain lacks labeled data entirely, making it crucial to assess the performance of MDTC approaches under such extreme conditions. In this particular setup, we are presented with multiple source domains containing both labeled and unlabeled samples, alongside a target domain that solely consists of unlabeled samples. Due to the absence of labeled data in the target domain, the evaluation protocol involves utilizing solely the shared features as input for both classifiers $\mathcal{C}$ and $\mathcal{C}'$, while setting the domain-specific vectors to be 0.

We conduct the multi-source unsupervised domain adaptation (MS-UDA) experiments on the Amazon review dataset. In this setting, three out of the four domains were treated as source domains, while the remaining one served as the target domain. We reported the classification results specifically for the target domain and compared them against various baselines. The baselines encompassed three domain-agnostic domain adaptation methods: (1) A multi-layer perceptron (MLP) model trained on the source domains, (2) the marginalized denoising autoencoder (mSDA) (Chen et al., 2012), and (3) the domain adversarial neural networks (DANN) (Ganin et al., 2016). For these three methods, it was necessary to combine all data from the source domains into a single domain. Moreover, we evaluated three state-of-the-art MS-UDA methods: (1) multi-source domain adaptation neural networks (MDAN) (Zhao et al., 2017), MAN (MAN-NLL and MAN-L2) (Chen & Cardie, 2018), and DACL (Wu & Guo, 2020). The MS-UDA results can be found in Table 6. Notably, our MDAT method outperformed all baselines across the four domains, surpassing the second-best approach DACL by a margin of 0.4% in terms of average classification accuracy. These results highlight the excellent generalization capabilities of our MDAT method in the MS-UDA setting.

## F LIMITATIONS

The primary contribution of this study lies in establishing a generalization bound for MDTC, bridging the gap between MDTC theories and algorithms. Although our proposed MDAT method may not surpass certain state-of-the-art MDTC methods in terms of the system performance, such as the maximum batch Frobenius norm (MBF) method (Wu et al., 2022b) and the co-regularized adversarial learning (CRAL) method (Wu et al., 2022a) (as illustrated in Table 7 and 8), it is important to note that our focus, as stated in Section 1 of the main content, is to provide theoretical guarantees for MDTC algorithm design. We achieve this by deriving a new generalization bound based on Rademacher complexity. Rather than relying on sophisticated architectural designs or a combination of modern techniques to achieve advancement, our contributions lie in developing concrete insights and theoretical foundations for MDTC.

Table 7: MDTC classification accuracies on the Amazon review dataset.

| Domain | MDAT(Proposed) | MBF | CRAL |
|--------|----------------|-------|-------|
| Books | $84.33 \pm 0.16$ | 84.58 | 85.26 |
| DVD | $86.07 \pm 0.11$ | 85.78 | 85.83 |
| Electr. | $88.74 \pm 0.06$ | 89.04 | 89.32 |
| Kit. | $90.56 \pm 0.13$ | 91.45 | 91.60 |
| AVG | $87.43 \pm 0.08$ | 87.71 | 88.00 |

Table 8: MDTC classification accuracies on the FDU-MTL dataset.

| Domain | MDAT(Proposed) | MBF | CRAL |
|--------|----------------|-------|-------|
| books | $88.1 \pm 0.2$ | 89.1 | 89.3 |
| electronics | $88.8 \pm 0.6$ | 91.0 | 89.1 |
| dvd | $91.0 \pm 0.4$ | 90.4 | 91.0 |
| kitchen | $92.1 \pm 0.4$ | 93.3 | 92.3 |
| apparel | $90.3 \pm 0.5$ | 88.5 | 91.6 |
| camera | $92.2 \pm 0.4$ | 92.8 | 96.3 |
| health | $89.8 \pm 0.2$ | 92.0 | 87.8 |
| music | $87.5 \pm 0.1$ | 85.9 | 88.1 |
| toys | $91.4 \pm 0.4$ | 92.2 | 91.6 |
| video | $90.7 \pm 0.6$ | 90.4 | 92.6 |
| baby | $90.0 \pm 0.3$ | 90.8 | 90.9 |
| magazine | $94.2 \pm 0.2$ | 93.5 | 95.2 |
| software | $91.3 \pm 0.5$ | 91.4 | 87.7 |
| sports | $89.7 \pm 0.3$ | 90.3 | 91.3 |
| IMDb | $89.4 \pm 0.1$ | 89.9 | 90.8 |
| MR | $77.0 \pm 0.5$ | 79.2 | 77.3 |
| AVG | $89.6 \pm 0.1$ | 90.1 | 90.2 |

