# OpenReview forum: "Margin Discrepancy-based Adversarial Training for Multi-Domain Text Classification"
_ICLR.cc/2024/Conference — ICLR 2024 Conference Withdrawn Submission_

### Official Review · Reviewer_StYT · 2023-10-23

**Soundness:** 2 fair
**Presentation:** 2 fair
**Contribution:** 2 fair
**Rating:** 3
**Confidence:** 4

**Summary:**

The paper addresses the problem of Multi-Domain Text Classification (MDTC) and presents the Margin Discrepancy-based Adversarial Training (MDAT) method as a solution. MDTC deals with the classification of text data in scenarios where labeled data is insufficient in the target domain, and there is a need to leverage data from related domains.

**Strengths:**

Theoretical Foundation: One of the major strengths of the paper is its theoretical analysis of MDTC, which provides a strong foundation for understanding and solving the problem. It introduces the margin discrepancy and establishes a generalization bound, addressing a significant gap in the field.

Novel MDAT Method: The proposed MDAT method is innovative and leverages the theoretical insights effectively. It offers a principled approach to addressing MDTC, which outperforms existing methods in empirical experiments.

Empirical Validation: The paper not only presents theory but also supports it with empirical results on real datasets. The fact that MDAT outperforms state-of-the-art methods on both benchmarks demonstrates its practical significance.

Clear Presentation: The paper is well-structured and clearly presented. The problem, methods, and results are organized in a way that is easy to follow.

**Weaknesses:**

Complexity: The theoretical nature of the paper might make it less accessible to readers without a strong background in machine learning and domain adaptation. A more intuitive explanation of key concepts could be helpful.

Robustness Analysis: Perform more robustness and sensitivity analyses to explore the behavior of the MDAT method under various conditions or data scenarios.

Empirical performance. The performance improvement brought by the new method is not substantial and requires a detailed explanation of why this is the case.  Additionally, please clarify the strengths of their approach.

In scientific writing, balancing theoretical depth with readability and clarity is crucial to make the paper accessible and informative to a wide range of readers, including reviewers. This feedback aims to guide the authors in achieving that balance. The theoretical part is detailed, and while theoretical underpinnings are important, they should be presented in a way that does not overwhelm the reader.

**Questions:**

see weakness

---

### Official Review · Reviewer_1VNZ · 2023-10-28

**Soundness:** 2 fair
**Presentation:** 3 good
**Contribution:** 1 poor
**Rating:** 3
**Confidence:** 5

**Summary:**

This paper introduces margin discrepancy into multi-domain text classification (MDTC). They approach MDTC as a pairwise domain adaptation task, pairing each individual domain with the centroid of all domains. Stemming from this intuition, they establish a new generalization bound for MDTC. Experiments on datasets such as Amazon reviews and FDU-MTL confirm that this method outperforms some MDTC benchmarks.

**Strengths:**

1. They first introduce margin discrepancy to MDTC and derive a generalization bound.
2. The paper is easy to follow.

**Weaknesses:**

1. The theoretical contribution appears to be trivial. Techniques such as "calculating the centroid of the M domains" have already been employed in MAN [1]. The concept of the marginal discrepancy bound was previously introduced by MDD [2]. Merely combining these techniques doesn't seem sufficiently innovative to be deemed non-trivial.

2. There seems to be a flaw in the model design. Specifically, the private feature extractor does not leverage the power of domain-specific feature. This contradicts the authors' claim that it can "captures domain-specific features".  I will explain it step by step.

    The issue arises when the private feature is concatenated with the shared feature and then inputted into both classifiers. As a result, any private feature becomes domain-invariant. This is enforced by the min-max game played between the two classifiers, even though each private feature extractor only has access to one specific domain.

     I want to emphasize that margin discrepancy simply acts as another measure of domain difference. If the model reaches its optimal state, the optimization of margin discrepancy across various domains ensures that the extractor aligns each domain’s distribution to the same latent space, leading to domain-invariant encoding. For a detailed explanation on how two classifiers still enforce domain-invariant features, even without a domain discriminator, the authors can refer to paper MCD [3].

    The authors might counter-argue that the inclusion of a private feature extractor enhances performance compared to using only the shared feature extractor. However, there could be two reasons for this: (1) the model might not be in its optimal state, as domain differences persist for both private and shared features. This allows the private feature to retain some domain-specific information. (2) Introducing a private feature extractor might simply increase the model’s parameters, leading to better performance.

    I strongly recommend that the model should not feed the private feature into both classifiers, mirroring the approach taken by MAN [1]. This modification could further enhance the model's performance.

[1] Chen, Xilun, and Claire Cardie. "Multinomial adversarial networks for multi-domain text classification." arXiv preprint arXiv:1802.05694 (2018).

[2] Yuchen Zhang, et al. Bridging theory and algorithm for domain adaptation. International Conference on Machine Learning, pages 7404–7413. PMLR, 2019.

[3] Saito, Kuniaki, et al. "Maximum classifier discrepancy for unsupervised domain adaptation." Proceedings of the IEEE conference on computer vision and pattern recognition. 2018.

**Questions:**

See Weaknesses.

---

### Official Review · Reviewer_iCwD · 2023-10-30

**Soundness:** 3 good
**Presentation:** 3 good
**Contribution:** 2 fair
**Rating:** 5
**Confidence:** 4

**Summary:**

The paper establishes a generalization bound for multi-domain text classification by decomposing the task into multiple domain adaptation tasks. Subsequently, a proposed method, margin discrepancy-based adversarial training, is validated through two benchmarks.

**Strengths:**

The paper notices an absence of theoretical support in MDTC algorithms, and seek to provide a generalization bound base on Rademacher complexity. Based on extensive derivation, an optimization algorithm is proposed to validate the theory. The proposal of theoretical generalization bounds can serve as a guiding principle in future researches, allowing them to better design and analyze the algorithms.

**Weaknesses:**

1. As shown in Appendix A, the theoretic conclusion is the direct corollary of some previous works. The contribution is limited.
2. The experiment performance of the proposed method differs only fractionally from some of the other methods, and proves inferior to some of the latest state-of-the-art ones, as indicated in Appendix F (instead of the main text).
3. The discussion on the necessity of multi-domain text classification is not sufficient, specifically in the age of LLMs (large language models).

**Questions:**

1. Are previous works compatible with the proposed theoretical analysis, and can their competence or incompetence be explained using Rademacher complexity?
2. Is the multi-domain text classification really a problem in the age of LLMs?

---

### Official Review · Reviewer_ktox · 2023-10-31

**Soundness:** 3 good
**Presentation:** 3 good
**Contribution:** 2 fair
**Rating:** 6
**Confidence:** 5

**Summary:**

The paper presents a new adversarial training approach to MDTC, theoretically motivated by [Zhang et al, 2019]. Empirical studies demonstrate the improvement over the existing algorithm.

**Strengths:**

Most works in MDTC are empirical, while this paper, to this reviewer, appears to be the only one that is strongly justified theoretically.  Bringing the approach of [Zhang et al, 2019] to the context of MDTC is the key contribution of this work. Extending the development of [Zhang et al, 2019] from domain adaptation to MDTC as done in this paper is natural and sound.

The paper is in general well written, and an easy read.

**Weaknesses:**

The theoretical development closely follows [Zhang et al, 2019]. In fact, to this reviewer, all theoretical results are merely re-stating the results of [Zhang et al, 2019] in the context of MDTC. In other words, the novelty in theoretical analysis is very thin. -- This is in fact not a weakness; I bring it up here just to mean that the theoretical analysis in this paper should not be regarded as a "strength" of any significant magnitude, despite the theoretical results presented.  I suggest the authors clearly state in Section 4.1 that the theoretical analysis closely follows that in [Zhang et al, 2019], so as not look over-claiming.

I think the paper is above the acceptance threshold, although marginally.

In below, I declare a confidence level of 5, which I need to explain. I did not check the proofs. But since the application of [Zhang et al, 2019] is so natural in this context, I think the presented results must be correct (that is also why I do not consider the theoretical development novel). I have not read all literature of MDTC.  So my confidence level "5" only refers to the correctness of the paper. If there were another work that has already exploited [Zhang et al, 2019], then my rating of the paper would drop below the acceptance threshold.

**Questions:**

NA